| Editor's Pick | Microbial Ecology | Research Article

# Nutrient limitation shapes functional traits of mycorrhizal fungi and phosphorus-cycling bacteria across an elevation gradient

Hannah B. Shulman,[1] Jessica A. M. Pyle,[1] Aimée T. Classen,[2] David W. Inouye,[3] Ruth Simberloff,[1] Patrick O. Sorensen,[4] William Thomas, IV,[1] Jennifer A. Rudgers,[5] Stephanie N. Kivlin[1]

**ABSTRACT** In nutrient-limited high-elevation ecosystems, plants rely on arbuscular mycorrhizal (AM) fungi to provide mineral phosphorus (P) in the form of phosphate ($PO_4^{3-}$). AM fungi gather these nutrients from phosphorus-cycling bacteria (PCBs) that can mineralize $PO_4^{3-}$ from organic matter and solubilize mineral-bound P. How climate, soil factors, and nutrient limitation influence AM fungi and PCB assembly remains unclear. We collected soil from montane meadows across a 1,000-m elevation gradient on three replicate mountainsides and analyzed AM fungal marker genes, P-cycling genes from shotgun metagenomes, and edaphic measurements. High-elevation soils had nearly 50-fold less soil $PO_4^{3-}$ and 60% more AM fungal hyphae than low-elevation soils. AM fungal turnover was linked to changes in pH, organic carbon, and $PO_4^{3-}$. The composition of 198 P-cycling genes was influenced by the AM fungal community structure. Drivers of individual PCB functional genes, including pH and organic carbon, varied with gene phylogeny. We found a trade-off in P-cycling strategies across elevation: P-rich, low-elevation soils supported root-colonizing AM fungi and organic P-mineralizing bacteria. P-poor, high-elevation soils were dominated by stress-tolerant AM fungi and mineral P-solubilizing bacteria. Our results suggest that AM fungi and PCB community turnover across elevation are both shaped by pH, organic carbon, and P availability. With continued climate warming, the structure and function of mountaintop ecosystems might shift to resemble lower elevations, disrupting long-established and specialized microbial assemblages, with consequences for P-cycling dynamics and the total P available to plant communities.

**IMPORTANCE** Phosphorus (P) limits plant productivity in high-elevation ecosystems, yet the microbial networks that mobilize P, including arbuscular mycorrhizal (AM) fungi and phosphorus-cycling bacteria (PCBs), remain under-characterized in these nutrient-poor soils. We show that across a 10,00-m elevation gradient, AM fungi and P-cycling gene assemblages shift predictably with pH, organic carbon, and phosphate availability. Higher elevations, with less available P, select for stress-tolerant AM fungal taxa and PCB strategies geared toward mineral solubilization, while low-elevation sites favor root colonization by AM fungi and organic P mineralization. These results suggest that nutrient limitation can constrain microbial community assembly in consistent ways across landscapes.

High mountain soils are low in P and rely on a network of underground AM fungi and PCB to deliver nutrients to plants. This study shows how those underground relationships reorganize with elevation and how climate change could collapse long-standing microbial strategies by pushing high-elevation ecosystems toward lowland conditions. As soils warm and dry, the microbial scaffolding that supports alpine plant life may become increasingly unstable.

**Peer Reviewer** Mia Rose Maltz, University of California Riverside, Riverside, California, USA

Address correspondence to Hannah B. Shulman, hshulman@utk.edu.

See the funding table on p. 13.

**KEYWORDS** arbuscular mycorrhizal fungi, montane ecosystem, phosphorus-cycling bacteria, metagenomics, genes-to-ecosystems, stress gradient hypothesis

The symbiosis between plants and arbuscular mycorrhizal (AM) fungi is fundamental for plant access to soil nutrients. AM fungi provide 70%–100% of plants' phosphorus (P) and nitrogen (N) in exchange for carbon (C) derived from plant photosynthesis [1, 2]. At the regional level, plant community composition and geography strongly influence AM fungal community composition, reflecting the effects of host plant, climate, and stochastic processes on shaping the AM fungal niche [3–7]. However, within ecosystems, abiotic soil factors like nutrient levels and pH may exert a stronger influence on the AM fungal community, as they directly affect the availability of resources and efficiency of biogeochemical processes that underlie fungal growth and function [8–10]. A mechanistic understanding of how nutrient limitation influences the composition and functional traits of the AM fungi may reveal their role in maintaining ecosystem stability and resilience under environmental stressors and improve predictions of ecosystem behavior under global change [9].

AM fungal functional trait frameworks explain what species-specific traits—such as physical structure, growth patterns, and nutrient limitation sensitivity—underlie changes in community composition [11–14]. A recently proposed trait framework [13] is based on resource allocation to different hyphal growth patterns, distinguishing AM fungal families by their investment in intraradical versus extraradical hyphae. Within this "ERA" framework for the Glomeromycota, Gigasporaceae and Diversisporaceae are classified as "*e*daphophilic," investing in high extraradical growth; Glomeraceae, Claroideoglomeraceae, and Paraglomeraceae are "*r*hizophilic," allocating more carbon to high intraradical growth; and Archaeosporaceae, Ambisporaceae, Acaulosporaceae, and Pacisporaceae are classified as "*a*ncestral," exhibiting both low intraradical and extraradical hyphal growth.

Different resource-based hyphal growth traits raise the question of how these functional traits are filtered along environmental gradients and whether the stress of nutrient limitation is related to specific AM fungal traits. In this study, we leverage the predictable patterns of edaphic and climatic variation across elevation in montane meadow ecosystems [15] as a model for studying how nutrient availability affects AM fungal trait patterns, diversity, and hyphal growth. At high elevations, short growing seasons, cooler temperatures, and higher rain and soil runoff have resulted in younger, energy-limited, and less-weathered soils that have lower nutrient availability and plant inputs [16, 17]. At low elevations, the longer growing season has resulted in more developed soils with higher nutrient concentrations that support more productive plant communities. Turnover of the microbial community across elevation is, therefore, often marked by decreasing species richness, organic matter cycling, and increasing abundance of oligotrophic lifestyles [18–20].

Environmental variation along elevation gradients offers a natural context for examining how mutualistic mycorrhizal relationships adapt to resource limitations. The stress gradient hypothesis (SGH) predicts that mutualistic interactions become more important in stressful environments where resource limitations constrain plant growth [21, 22]. In the context of AM fungi in montane ecosystems, the SGH suggests a shift from competition to facilitation between AM fungi and their plant partners as nutrient availability decreases with elevation [23] increasing the benefit gained from AM fungal symbiosis [24]. These high-elevation ecosystems and the mutualisms that support them are also vulnerable to climate change [25]. Communities and processes at higher elevations may increasingly resemble those at lower elevations as mean annual temperatures rise (26; Souza et al., unpublished data). A better understanding of how nutrient limitation selects for AM fungal traits will therefore provide mechanistic insight into the maintenance of mutualistic relationships under future global change scenarios. Gradient studies may also better estimate the effects of shifting climates than experi-

mental manipulations which, due to their limited time span and spatial imprint, can underpredict effects on plants and their microbial symbionts (15, 27, 28).

At higher elevations, soil available P is lower due to a combination of P loss through erosion and high microbial P demand (29–32). AM fungal hyphae can absorb this P from different soil pools, including mineral P that can be solubilized with organic acids (33, 34) and organic P, which exists most abundantly in soil in the form of phosphate esters like inositol phosphate (35). AM fungi cooperate with P-cycling bacteria (PCBs) to mineralize and gather P from the soil (36). These co-operating bacteria can grow inside the AM fungi, on the surface of their hyphae, or in the bulk soil where they are stimulated by hyphal exudates to increase decomposition and P mineralization (37–40). However, the degree to which AM fungi directly mineralize P through acid phosphatase secretion (41, 42) versus rely on their associated microbial partners may vary across taxa and environments, potentially leading to differential trait-based responses to P limitation across the landscape (38–40).

Here, we investigate changes in the composition and functional traits of both AM fungi and PCBs over elevation to determine how resource availability is related to the assembly of this P-cycling microbial consortium. We studied AM fungi and PCBs from 2,700 to 3,700 m in elevation in montane meadow soils of the Colorado Rocky Mountains. We generated a data set of AM fungal SSU marker genes, soil bacterial P-cycling genes, and climate/edaphic measurements to determine how community dynamics shift with elevation and changing P concentrations, testing the following hypotheses.

   i. As elevation increases, declining soil nutrients will result in more-developed AM fungal hyphae and a shift in AM fungal community composition, reflecting increased investment of plants into AM fungal nutrient foraging processes. We expected that hyphal growth traits would influence AM fungal composition, favoring rhizophilic (root-colonizing) fungi at lower elevations and edaphophilic (soil-colonizing) fungi at higher elevations.
   ii. Edaphic changes with elevation change the functional inventory of PCBs. We expected a higher prevalence of genes related to particulate, enzymatically derived P, such as acid phosphatase, in soils at low elevations and mineral-associated, chemically derived P-cycling genes, such as gluconic acid producing genes, in soils at high elevations.

If supported, these results may shed light on the functional linkages between mycorrhizal fungi and bacteria at the ecosystem scale and provide mechanistic insight into how P-cycling assemblages adapt to the stress of nutrient limitation.

## MATERIALS AND METHODS

### Field system and experimental design

We sampled six locations across 1,000 m of elevation on three distinct focal mountain peaks (Avery, Cinnamon, and Treasury) in the Colorado Rocky Mountains (separated by a maximum of 11 km). We collected soils from 0 to 10 cm depth using a hand trowel at each of the six elevations that were spaced 100–200 m apart (2,700–3,700 m, spanning from montane to alpine ecosystems) weekly for 5 weeks during peak growing season (1 July to 6 August) in 2018. At each elevation on each of these three mountains, we collected soil from 10 subplots located along a 100-m transect and homogenized them into one sample to account for microsite variability. We collected a total of 90 soil samples.

Adiabatic lapse rate changes temperature across this gradient by ~0.8°C per 100 m (43), which is analogous to ~10 years of climate change in the southern Rockies. The plant communities were a mix of perennial grasses and forbs (e.g., *Achnatherum lettermanii*, *Festuca thurberi*, *Poa pratensis*, and *Elymus trachycaulus*). Soil textures at all sites are clay loam to sandy clay loam with an average bulk density of 0.91 (±0.21) g/cm$^3$.

Mean temperatures vary by ~3°C between the lowest and highest elevation sites on each gradient (44). Soils were processed the same day as collection and refrigerated until biogeochemistry assays were completed, with a subset frozen immediately at −80°C for DNA extraction. We completed microbial biomass and soil nutrient extractions within 24 h of sample collection.

## Soil biogeochemistry

We measured gravimetric soil moisture (105°C for 48 h) and determined soil organic matter by loss on ignition (360°C for 2 h) on ~5 g soil for each assay (45). We converted organic matter content to organic C content using a correction factor of 0.284 (46). Soil pH was measured in DI water (1:2, wt/vol) that was homogenized on a reciprocal shaker for 1 h. We extracted and quantified $NH_4^+$ and $PO_4^{3-}$ using standard methods (47, 48). To assess fungal abundance in soil, we quantified fungal hyphal length microscopically. We extracted fungal hyphae from field-moist soil stored at 4°C within 2 weeks of sample collection using a sodium hexametaphosphate flocculant. We used the gridline-intercept method on a compound light microscope at 200× magnification to quantify aseptate (AM fungal) hyphal length (49).

We measured microbial biomass C (MBC) from 5 g of soil using the fumigation-extraction method (50, 51). We shook control and chloroform-fumigated soil in 0.5 M $K_2SO_4$, and then, extracts were frozen until later quantification. Extracts were analyzed colorimetrically after mixing with 10 mM Mn(III)-pyrophosphate and 98% $H_2SO_4$ and incubated in the dark at room temperature for 16 h. We quantified absorbance on a spectrophotometer at 495 nm (52). We measured acid phosphatase activity using the 4-methylumbelliferyl phosphate substrate enzyme assay (53).

## Sequencing and bioinformatics

We extracted DNA from soils using the Qiagen DNEasy Soil Kit. We characterized AM fungal communities by amplifying the small subunit rDNA using NS31/AML2 primers (54) and sequencing on the Illumina MiSeq with 2 × 300 bp reads. ASVs were classified using QIIME2 and the MaarjAM database (3). AM fungal ASVs were also classified into the ERA (edaphophilic, rhizophilic, or ancestral) guild framework for hyphal growth patterns (13).

Shotgun metagenomes were sequenced and analyzed by the Joint Genome Institute on an Illumina MiSeq (2 × 150 bp reads). Contigs were assembled with metaSPAdes and annotated using the KEGG, Pfam, Tiger, and FunFam databases. We pulled all known, important terrestrial P-cycling genes from our data set based on PcycDB (55). In addition to known P-cycling genes, we pulled all putative P-cycling genes encoding proteins that act on phospho-ester, phosphoric acid anhydride, or phosphonate bonds and those that produce phospho-ester products based on Enzyme Commission annotations. To filter out basal cellular processes that are unlikely to affect ecosystem level P-cycling, we analyzed the putative p-cycling genes with SignalP 6.0 (56) and manually curated the recovered genes to determine which genes produced enzymes that would be secreted to the extracellular soil matrix. The taxonomic breakdown of P-cycling genes was 98.4% bacteria, 0.33% archaea, 0.2% fungi, 0.24% other eukaryotes, and 0.84% unassigned; therefore, we constrained the functional gene analysis to bacteria only. All P-cycling genes found in the shotgun metagenome data set were grouped into functional trait categories based on KEGG metabolic pathways (Table S4). We normalized all gene abundances to the abundance of the single-copy, widely conserved, "internal housekeeping" gene rpoB (57) to control for differences in read depth across samples. When analyzing the community composition of PCBs, contigs were summarized by both identity of functional gene on the contig and species identity of the contig (e.g., see Table S2: PCBs "by function" vs "by species"). For full details of our marker gene and metagenomic bioinformatic pipelines and supplemental data, please refer to the supplemental material (bioinformatics appendix).

We recovered 198 unique gene products related to phosphorus cycling, occurring in 22,391 contiguous sequences and 3,718 bacterial taxa in our data set. We identified

35 well-characterized genes described in PcycDB, accounting for 85% of the P-cycling contigs and 80% of total P-cycling read depth. Signal peptide analysis with SignalP identified an additional 179 genes encoding putative exoenzymes: 64% hydrolases, 12% transferases acting on organophosphorus compounds, 14% poorly characterized hypothetical proteins with phosphorus-transforming domains, and 10% other enzyme classes. Most PCB genes were mapped to Proteobacteria, Actinobacteria, or Acidobacteria lineages (Fig. S2). The most abundant pathways were phosphorus solubilization, extracellular phospholipid turnover, and extracellular phosphatases that mineralize $PO_4^{3-}$ from organic matter (Fig. S2).

## Statistical analysis

All statistical analyses were performed in R. We ran mixed-effects linear models using the nlme package to determine fixed effects (elevation, pH, and other edaphic measurements: %SOC, $NH_4^+$, $PO_4^{3-}$, MCB, and AP) on responses of AM fungal hyphal length, the abundance of P-cycling gene groups, and microbial Shannon diversity. We determined that there was no collinearity of our fixed effects with the vif function. We verified that model assumptions were met by inspecting residual–fitted and Q–Q plots and confirming that residuals were approximately normally distributed and homoscedastic, supporting the use of linear models. The replicate transects from each mountain ($n$ = 3: Avery, Treasury, and Cinnamon) were used as a random effect. We accounted for first-order temporal autocorrelation based on sampling week using the corARMA function. We calculated adjusted $R^2$ for our models with the r.squaredGLMM function. We determined effect sizes using standardized beta (β) coefficients calculated with the standardize_parameters function.

We performed ordinations and variance partitioning to determine the contribution of these fixed effects on AM fungal and PCB community composition. Both AM fungal composition and PCB composition were center-log normalized (58, 59), and distance matrices were calculated for each with Aitchison distance (60). We ran dbRDA models using the R function dbrda to determine the effects of climate and edaphic effects on microbial composition, using stepwise regression (stepAIC in R, both directions) to find the best-fitting models. To determine the interacting effects of PCB and AM fungal composition, principal coordinate axes from unconstrained ordinations were incorporated into dbRDA models (i.e., unconstrained principal component axes for AM fungi were used as fixed effects in models for PCB composition). Models were conditioned on both sampling week and replicate gradient. To determine the percent of community variation driven by the significant effects, we ran variance partitioning using the varpart function in vegan.

To construct an AM fungal-PCB network, we calculated Kendall rank correlation coefficient among AM fungal ASVs binned at the virtual taxonomic unit level and PCB genes binned at the phylum level on each mountain across elevations (18 total networks). We made networks out of all significant correlations with Kendall ranks >0.88 in R using igraph (61). We used igraph to quantify the number of AM fungal taxa and PCB genes with significant correlations (vertices/nodes) and the number of correlations between vertices (edges/connections).

## RESULTS

### Soil pH, AM fungal hyphal length, and nutrient limitation across elevation

As expected, with increasing elevation, AM fungal hyphae increased in length by 1.5 m per gram of soil from 2,700 to 3,500 m (β = 0.35; Fig. 1A; Table 1). We found that higher elevations were consistently lower in $PO_4^{3-}$, which dropped by 50-fold from low- to high-elevation sites (β = −0.91; Fig. 1B). $NH_4^+$ varied from 0.5 to 80 grams per gram of soil but did not have consistent elevation or pH-dependent trends (Fig. 1D; Table 1). Instead, $NH_4^+$ levels were most variable by sampling date, increasing across the 5-week sampling duration ($F[1,49]$ = 169, $P < 0.001$). Neither $PO_4^{3-}$ nor hyphal lengths changed

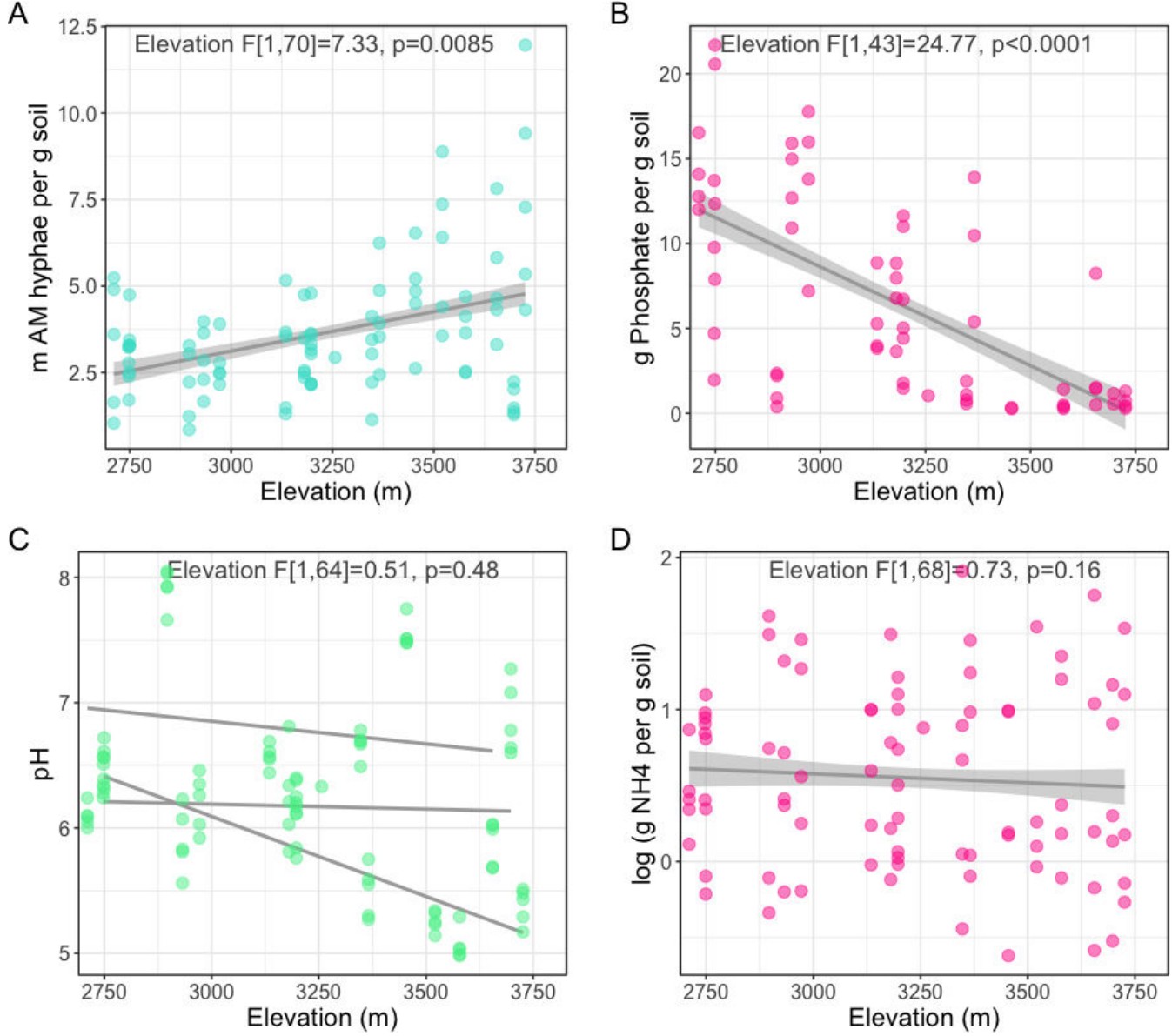

**FIG 1** High-elevation soil has less $PO_4^{3-}$ and longer AM fungal hyphae. Linear response of AM fungal hyphal length (A), $PO_4^{3-}$ (B), pH (C), and $NH_4$ (D) to elevation. All measurements were sampled across approximately 1,000 m of elevation on three distinct mountains, which are shown as trendlines in plot C to emphasize spatial variation in pH dynamics. The F-means and significance of elevation effect from the mixed-effects linear model are shown in the top right corner.

across time. Soil pH was negatively correlated with hyphal lengths ($\beta = -0.26$) and $PO_4^{3-}$ ($\beta = -0.34$) but did not exhibit a consistent trend across elevation (Fig. 1C), suggesting that the effects of pH on hyphal growth and P limitation are related to site specificity and local adaptation.

Both SOC and MBC were measured to determine the impacts of organic matter on AM fungal dynamics. With increasing elevation, MBC decreased and was negatively correlated with hyphal lengths ($\beta = -0.18$; Fig. S1G). SOC varied from 1% to 4% without clear elevation-based or pH-dependent trends (Fig. S1E and F). Acid phosphatase activity was higher in more acidic soil ($\beta = -0.52$) but was not correlated to $PO_4^{3-}$ concentrations or AM fungal hyphal lengths (Fig. S1J).

**TABLE 1** Models of soil nutrients and AM growth across elevation[a,b]

| Response | pH | Elevation | AM HL | Lat | Long | denDF | Intercept |
|---|---|---|---|---|---|---|---|
| AM HL | 4.94* | 7.33** | – | – | 1.99 | 70 | 173.49 |
| $[PO_4^{3-}]$ | 5.58* | 24.77*** | – | – | – | 43 | 66.65 |
| $[NH_4]$ | – | 0.73 | – | 1.42 | – | 68 | 54.04 |
| %SOC | – | – | – | – | – | 71 | 207.4 |
| MBC | – | – | 4.97* | – | – | 51 | 65.35 |
| AP | 6.66* | – | 0.48 | – | 4.55* | 65 | 384.97 |

[a]Results from six mixed-effects linear models showing the effects of pH, elevation, AM hyphal length, and location on soil properties Cells show the $F$-values from type 3 ANOVA with asterisks indicating significance level (*<0.05, **<0.01, ***<0.001). Random effects: ~1|replicate mountain/plot. Temporal autocorrelation structure: corARMA (form = ~day of year|replicate mountain/plot, $P = 1$, $q = 0$). Cells without values (–) indicate that effect was dropped during model selection.
[b] AM HL = arbuscular mycorrhizal hyphal length, MCB = microbial biomass carbon, AP = acid phosphatase, Lat = latitude, Long = longitude.

## Composition of AM fungi and PCB

We ran dbRDA models with *post hoc* variance partitioning on both AM fungal marker gene communities and PCB functional gene communities in order to determine what soil nutrients shape composition over elevation (Fig. 2; Table S1).

AM fungal community composition was most strongly related to soil pH, which explained 4.9% of variation (Table S1; Fig. 2C). Across elevation, AM fungal turnover was related to the edaphic factors SOC and $PO_4^{3-}$ (Fig. 2A), although the unique effect of $PO_4^{3-}$ on the AM fungal community was small (0.6%). The composition of PCB species and AM fungi was significantly correlated, with the PCB community explaining 2.8% of AM fungal community composition (Fig. 2A and C; Table S1).

PCB species composition was most strongly related to AM fungal composition, which had a unique effect of 3.8% and a combined effect >15% (Table S1; Fig. 2D). Compared to variance partitioning of PCB functional composition based on edaphic factors alone, the addition of AM fungal principal coordinate axes increased explanatory power by 4.8%. The unique effects of elevation, pH, and edaphic factors were each <1%, with most of the variance explained combined with the AM fungal community. Ordination of PCB composition shows that this influence of AM fungi on PCBs is tightly correlated with SOC (Fig. 2B).

Our separate analyses of PCBs by species and gene function were fruitful, revealing that PCB composition by species is related to AM fungi but composition by gene product/function is not (Table S1). These results suggest that fungal-driven shifts in PCB composition may be functionally redundant, related to niche structuring, and represent local co-adaptation of bacteria and mycorrhizal fungi to soil conditions like soil carbon dynamics. Last, we found that AM fungal composition was not influenced by overall bacterial community structure (rpoB) or vice versa (Table S1), indicating that AM fungi have a unique effect on the PCB community.

## Functional traits of AM fungi and PCB

We found that 87% of the recovered P-cycling genes belonged to three functional groups (extracellular phosphatases, extracellular phospholipid turnover, and phosphorus solubilization via gluconic acid secretion, Fig. 3). Furthermore, 81% of the contigs mapped to three phyla (Actinobacteria, Proteobacteria, and Acidobacteria). We ran separate mixed-effects models on these key groups in addition to all PCBs to understand how edaphic factors influence P-cycling function over elevation (Fig. 3, Table S2).

With increasing elevation, Actinobacteria decreased in abundance (β = −0.27), while Acidobacteria and Proteobacteria increased (Acido β = 0.42, Prot β = 0.21, Fig. 3E). This taxonomic turnover mirrored the change in functional genes across elevation: with increasing elevation, phosphate solubilization genes increase (Acido β = 0.35, Prot β = 0.26; Fig. 3B) and extracellular phospholipid turnover genes decrease (Actino β = −0.33;

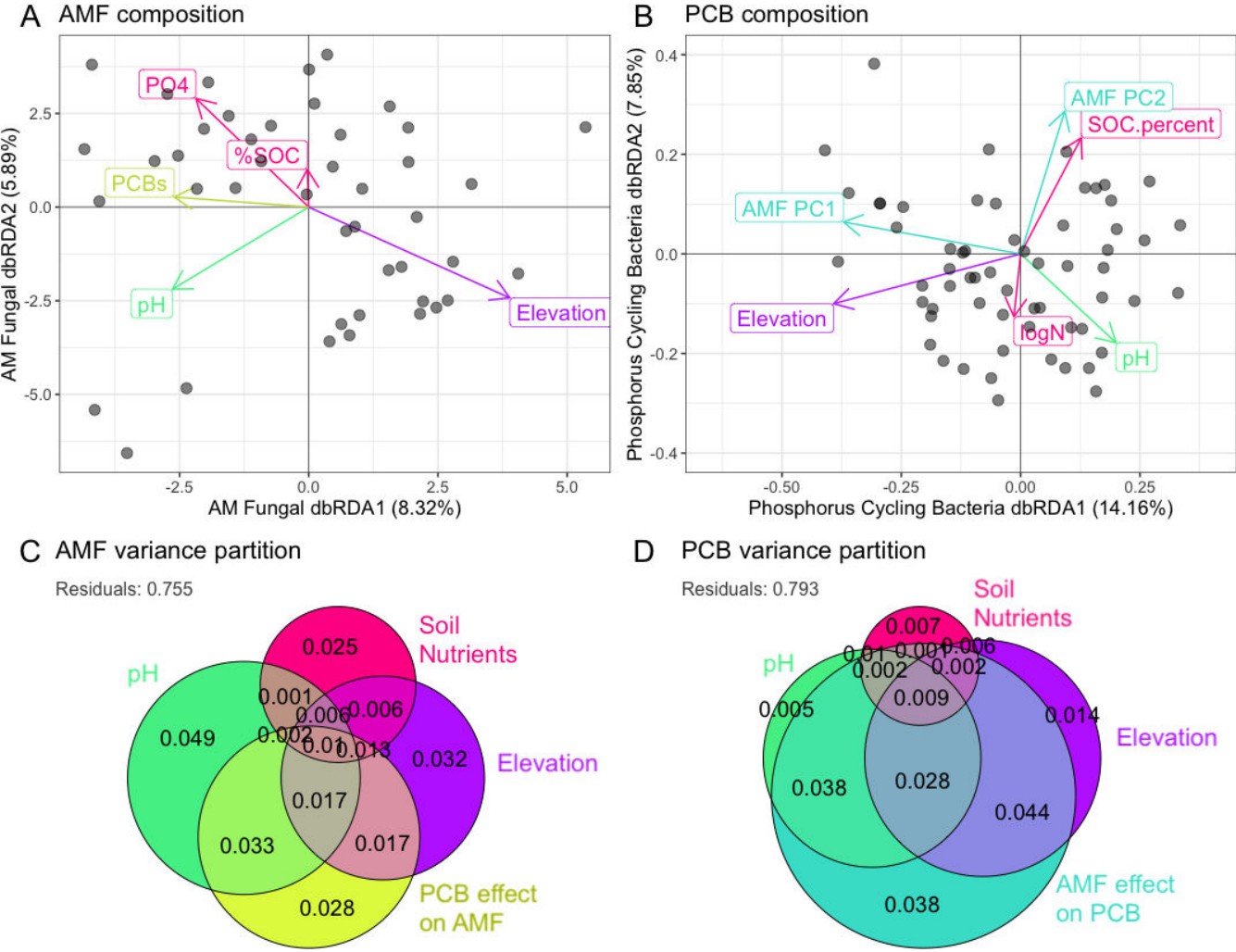

**FIG 2** Soil pH and nutrients shape assembly of AM fungi and PCBs. Constrained ordination on the taxonomic composition of AM fungi (A) and PCBs (B), with arrows showing significant environmental effects determined with dbRDA. Variance partitioning of AM fungi (C) and PCBs (D) is shown with euler plots. Plots are labeled with partial and combined percent of community variation driven by each significant factor, and circles are sized by total variance driven by each factor.

Fig. 3C). The effects of pH and %SOC on these functional trait patterns varied by phylum (Table S2), suggesting that community filtering differed between P-cycling clades.

We also found that $PO_4^{3-}$ concentrations were negatively correlated with both the functional diversity of the PCB community ($\beta = -0.07$) and the abundance of phosphate solubilization genes (Acido $\beta = -0.10$, Prot $\beta = -0.18$, Table S2). These results support that increasing P limitation at higher elevations is correlated with changes in composition and function of PCBs. Overall, these results show that in low-elevation, high $[PO_4^{3-}]$ soils, the PCB community is dominated by Actinobacteria capable of mineralizing P from organic matter. As elevation increases and $[PO_4^{3-}]$ drops, Proteobacteria and Acidobacteria capable of solubilizing phosphate from mineral surfaces via gluconic acid secretion increase in abundance.

We found trade-offs in the abundance and diversity of different AM fungal functional groups across elevations (Fig. 4). With increasing elevation, the relative abundance and diversity of the ancestral AM fungi, notably the Acaulospora and Archaeospora, increased by almost 200% (Fig. 4A; Fig. S3and Table S3). Very few ancestral AM fungal ASVs were found below 3,000 m. Ancestral AM diversity was significantly correlated to $PO_4^{3-}$ concentrations ($\beta = -0.12$), pH ($\beta = -0.22$), and %SOC ($\beta = 0.32$) across elevation. Rhizophilic diversity displayed the opposite pattern, decreasing with elevation ($\beta =$

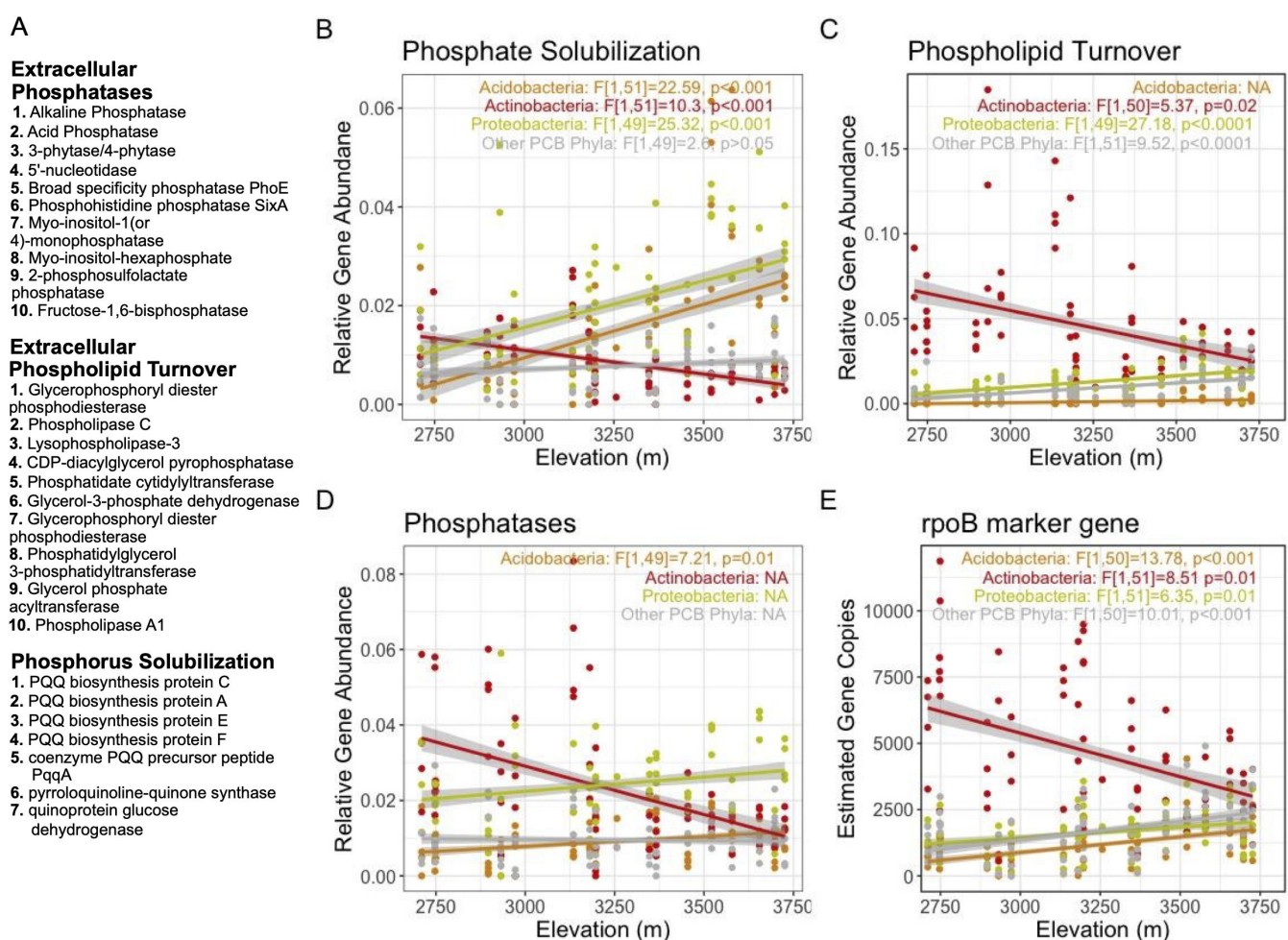

**FIG 3** Abundance of PCB genes changes across elevation. Patterns of bacterial gene abundance over elevation are shown for three different phosphorus-cycling functional groups (B–D). The 10 most abundant functional genes in each group are listed (A). Linear trends are shown separately for Acidobacteria, Actinobacteria, and Proteobacteria phyla by color. Marker gene rpoB shows general taxonomic abundance patterns over elevation (E). *F*- and *P*-values for elevation from the mixed-effect linear models are shown for each phyla. NAs indicate that elevation was dropped during model selection. For rpoB, gene copies are quantified by the calculated estimated gene copy number from the bioinformatics pipeline. For other functional genes, estimated gene copies were normalized to rpoB.

−0.22). This group was still the most abundant and diverse AM fungal group at all sites, especially the *Glomus* genus, although bias toward *Glomus* is expected with this primer set (62). We did not see clear patterns of edaphophilic AM fungi (most abundant genus: *Diversispora*) related to elevation or edaphic factors (Fig. 4A; Fig. S3).

To investigate how shifts in AM fungi relate to their ecological interactions with PCBs across elevation, we constructed co-occurrence networks linking AM fungal functional groups with PCB taxa. As ancestral AM fungi diversified with increasing elevation and decreasing [$PO_4^{3-}$], we also observed a larger number of ancestral AM fungi co-occurring with PCBs (β = 0.51, Fig. 4B). Likewise, as rhizophilic fungi decreased in diversity with increasing elevation, fewer species co-occur with PCBs (β = −0.70). However, we did not detect a clear pattern of the number of total AM fungal-PCB network interactions or changing network patterns of PCB genes across elevation. These results indicate that elevational shifts in AM fungal diversity and nutrient availability are related to the function, composition, and structure of joint fungal-bacterial communities.

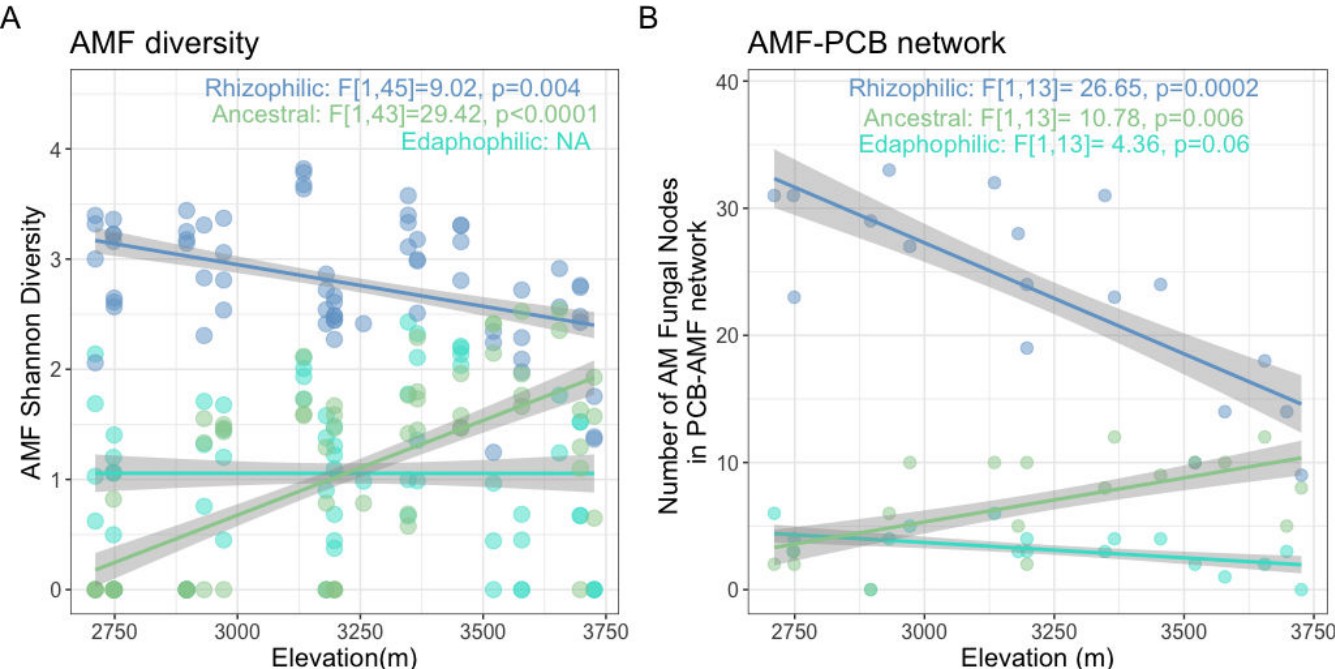

**FIG 4** AM fungal functional traits influence the community structure over elevation. (A) The linear trend of Shannon diversity vs elevation is shown for each AM fungal guild with the mixed-effects linear model results. (B) The number of network connections between AM fungi and PCB was calculated by creating Kendall rank correlation networks. The linear trend of network nodes abundance vs elevation is shown for each AM fungal guild with the mixed-effects linear model results.

## DISCUSSION

### Hyphal growth traits change as AM fungal community shifts over elevation

We found that rhizophilic fungi (*Glomus*, *Claroideoglomus*, *Rhizophagus*, and *Septoglomus* spp.) dominate the AM fungal community at low elevations and that low-elevation soil is colonized by less hyphae. Hyphal density increased and the AM fungal community shifted with increasing elevation and lower $PO_4^{3-}$, reflecting the influence of nutrient limitation on the fungal community and hyphal growth strategies. These results support hypothesis i and support that plant investment into AM fungi under P-limitation stress structures mycorrhizal communities at the ecosystem scale (63). With increasing elevation as P becomes limited, community composition shifts toward ancestral AM fungi, such as *Acaulospora* and *Archaeospora*. These results are contrary to our prediction that edaphophilic AM fungi would be more abundant with increasing elevation and nutrient limitation. While we were not able to determine species-specific hyphal growth patterns, the taxonomically based functional groups of the ERA framework (13) and hyphal data taken together show that hyphal allocation traits track our measurements of hyphal length density in bulk soil. Rhizophilic AM fungi, which preferentially allocate growth to the intraradical space, were associated with less hyphal colonization of bulk soil and less P limitation. Ancestral AM fungi were correlated with more total hyphal colonization of the soil and high P limitation. This pattern supports that ample P availability reduces selective pressure for extraradical foraging, leading to lower investment in soil‑exploring hyphae. Our results support other findings that ancestral AM fungi are better equipped to persist and thrive in stressful, resource-limited environments like mountaintops and have strong responses to P limitation (11, 64–67). However, it seems contradictory that soils with more abundant ancestral AM fungi like *Acaulospora* and *Archaeospora* also have more hyphae, as this ancestral type is associated with low rates of hyphal growth (13, 68). Based on our observations, it is possible that ancestral hyphae are longer lived, and a key mechanism of their stress tolerance could

be decomposing or otherwise turning over more slowly than other mycorrhizal types, demanding less C from host plants and returning more P per C in resource-limited environments. Rhizophilic AM fungal taxa like *Glomus* and *Rhizophagus* have high hyphal turnover rates, fast growth rates, and cell walls made up of more acid-hydrolysable compounds compared to other fungi, which could result in less hyphae being observed in the soil due to faster decomposition (11, 69). Future research focusing on species-specific differences in AM fungal turnover and hyphal chemistry, for example, C to N ratios, would improve understanding of different strategies for regulating host C supply as it has for ectomycorrhizae and saprotrophs (70, 71). Another possible explanation for why more P-limited, higher elevation soils have more hyphae and more ancestral-type AM fungi may be related to our observation that rhizophilic diversity is still relatively high at high elevations (Fig. 4). With increased niche space for stress-tolerant ancestral AM fungi in high-elevation, nutrient-poor soils, there could be complementary effects between rhizophilic and ancestral types that result in more P mineralization and total soil hyphae (72, 73). While elevational shifts in AM fungal communities suggest ecological strategies tied to nutrient limitation, it remains unclear how species-specific hyphal traits and interspecies interactions contribute to soil hyphal abundance and P cycling.

## Co-adaptation of P-cycling microbial consortium to P limitation

Recent development of research on hyphosphere and plant-fungal-bacterial tripartite symbiosis has created a flush of evidence supporting that AM fungal-bacterial relationships are important for provisioning plant nutrients (74–77). Phosphorus availability regulates AM fungal interactions with bacteria, which may allocate more plant-derived C to PCB when P is limited (78). Those bacteria, in turn, can mobilize soil organic P and mineral P in response to AM fungal priming (40, 79), increasing P cycling in response to P limitation (80). AM fungi can also impact the diversity and composition of bulk soil bacterial communities, which leads to increased P uptake by plants and faster litter decomposition (81, 82).

This study examined how communities of PCBs and AM fungi vary across ecosystems that differ in nutrient availability. Our results present evidence that the composition and specific functions of the AM fungal community are related to the taxonomic composition of the PCB community at ecosystem scales and may shape the PCB niche along with abiotic drivers like soil pH and organic carbon content. These results support similar findings that the fungal community has larger predictive power over the bacterial community than vice versa (83). We also observed that the diversity of AM fungal functional groups is positively related to their interactions with PCBs, suggesting that greater functional diversity among AM fungi may shift bacterial community assembly for enhanced P-cycling capacity. This may occur if diversity in hyphal growth strategies creates a broader range of nutrient niches and soil microhabitats, fostering complementary PCB functions and enhancing overall phosphorus turnover.

Our findings build on growing evidence that AM fungi actively shape bacterial communities involved in P cycling, reinforcing the importance of fungal-bacterial cooperation in nutrient-limited ecosystems. However, a key limitation of this study is that we did not directly isolate or characterize hyphosphere bacterial communities, which limits our ability to identify direct AM fungal-mediated bacterial recruitment or activity. Future AM fungal trait research that incorporates hyphosphere dynamics, and future hyphosphere research that incorporates fungal species-specific effects, will enhance understanding of microbial interactions and ecosystem-scale nutrient dynamics.

## A functionally and taxonomically diverse consortium of bacterial genes transforms phosphorus in montane soils

Here, we leveraged shotgun metagenomes as records of microbial life-history strategies to link functional potential with community composition (84). This approach allowed us to identify key phosphorus-cycling functions that vary at the landscape scale. The functional and species composition of PCBs shifted across elevation, correlated to AM

fungal community changes and soil nutrients. At low elevations, the P-cycling gene inventory was dominated by actinobacterial phospholipid turnover. At high elevations, the P-cycling gene inventory was dominated by P solubilization through gluconic acid secretion by Acidobacteria and Proteobacteria. This shift aligns with frameworks for terrestrial P limitation (85). High-elevation soils are less developed and more weathered, with less of a physical soil barrier that limits microbial access to mineral P. Low-elevation soils have higher plant productivity, higher litter inputs, and more P-rich particulate organic matter (31). It follows, therefore, that communities at low elevation have more genes that break down phospho-organic matter, and high elevations have a larger genetic inventory for solubilizing mineral P. At high elevations, where both AM fungal hyphae and PCBs can access mineral phosphorus sources more readily, the physical hyphal exploration of the soil matrix may also allow PCBs to access more mineral surfaces (86, 87).

The high abundance of genes encoding exoenzymes that break down phospholipids, such as phosphodiesterase and phospholipase, suggests that lipid membrane decomposition may be a key source of P in more-developed, nutrient-rich, low-elevation soils high in microbial biomass (88–90). Extracellular lipase enzymes have been identified in diverse soil bacteria (91, 92). Lipase enzyme activity in soil has been linked to hydric stress responses and organic matter turnover (93, 94). Our findings are similar to another study showing that lipase-producing bacteria increase in diversity at lower elevations in the Tibetan Plateau region (95). A large gene inventory for microbial biomass turnover at low elevation supports other findings that there is higher microbial turnover in warmer soil, that microbial biomass breaks down less readily at higher elevation, and that soil mineralogy strongly affects microbial biomass accumulation (96, 97). Here, we show that phospholipid-specific lipases may be a key bacterial trait that shapes ecosystem capacity for P cycling.

These results partially support hypothesis ii.While we did observe more P solubilization genes at higher elevations, we did not see a larger or more diverse inventory for extracellular phosphatases at lower elevations. Genes for these phosphate-releasing hydrolase enzymes, including phytase, alkaline phosphatase, and acid phosphatase, were among the most abundant P-cycling genes found in this transect but not related to $PO_4^{3-}$ concentrations and only affected by elevation in the Actinobacteria. We propose that promiscuous phosphatases (98), encoded by most, if not all, PCB genomes, serve as the "downstream" step for releasing $PO_4^{3-}$ in a pathway of phosphorus-transforming reactions. Our results suggest that it is these "upstream" P-cycling genes that vary among PCB taxa and with substrate availability or environmental conditions.

## Stress-gradient hypothesis

In this study, we surveyed landscape-level patterns of AM fungal and PCB composition to determine how soil nutrients, pH, and elevation are related to the assembly of this P-cycling consortium. Our results support the SGH (22, 23, 99), as we observed longer nutrient-foraging hyphae and more network linkages between PCBs and ancestral AM fungi at higher, P-limited elevations. Our results also show that soil P-cycling communities are not functionally redundant across elevation. We found that key microbial groups associated with stress tolerance increase with elevation and nutrient limitation, including the ancestral AM fungi and Acidobacteria, which are adapted to oligotrophic conditions (100). These patterns likely reflect both increased microbial cooperation under P limitation and environmental filtering that favors stress-adapted taxa in low-nutrient soils. Given that the SGH was originally developed for plants (21) and later extended to plant-soil feedbacks (101), further work is needed to determine whether these patterns hold broadly for microbial nutrient cycling communities.

Ongoing warming could ease P limitation at high elevations by increasing plant productivity and litter inputs (102, 103), accelerating microbial phospholipid turnover, and shifting P acquisition from inorganic to organic sources. Additionally, if warming weakens AM fungal associations (104), fungal-PCB interactions may also diminish,

altering P-cycling dynamics. These changes could cause high-elevation PCB communities to resemble those at lower elevations, transforming the stress-adapted P-cycling community into a "no-analog ecosystem" with reduced resilience to increasing climatic perturbations (105–108). Overall, our findings highlight key inter-kingdom dependencies among AM fungi and PCBs across environmental gradients, advancing our understanding of P-cycling mechanisms in montane ecosystems threatened by global change

## ACKNOWLEDGMENTS

S.N.K. was supported by NSF grants DEB2217353, DEB2106065, and DEB1936195. This research was also funded by the U.S. Department of Energy, Office of Science, Office of Biological and Environmental Research, Terrestrial Ecosystem Sciences program under the award number DE-FOA-0002392.

These metagenome data were generated for JGI Proposal #504086.

## AUTHOR AFFILIATIONS

[1]Department of Ecology and Evolutionary Biology, University of Tennessee Knoxville, Knoxville, Tennessee, USA

[2]Department of Ecology and Evolutionary Biology, University of Michigan, Ann Arbor, Michigan, USA

[3]Department of Biology, University of Maryland, College Park, Maryland, USA

[4]Division of Earth and Environmental Sciences, Lawrence Berkeley National Laboratory, Berkeley, California, USA

[5]Department of Biology, University of New Mexico, Albuquerque, New Mexico, USA

## AUTHOR ORCIDs

Hannah B. Shulman  http://orcid.org/0000-0002-9959-9417
Patrick O. Sorensen  http://orcid.org/0000-0002-0558-2789

## FUNDING

| Funder | Grant(s) | Author(s) |
|---|---|---|
| U.S. Department of Energy | DE-FOA-0002392 | Aimée T. Classen |
| | | David W. Inouye |
| | | Patrick O. Sorensen |
| | | Stephanie N. Kivlin |
| National Science Foundation | DEB2217353, DEB2106065, DEB1936195, DEB2338421 | Hannah B. Shulman |
| | | Aimée T. Classen |
| | | Jennifer A. Rudgers |
| | | Stephanie N. Kivlin |
| Joint Genome Institute | 504086 | Hannah B. Shulman |
| | | Stephanie N. Kivlin |

## AUTHOR CONTRIBUTIONS

Hannah B. Shulman, Conceptualization, Data curation, Formal analysis, Investigation, Validation, Visualization, Writing – original draft, Writing – review and editing | Jessica A. M. Pyle, Investigation, Methodology, Writing – original draft, Writing – review and editing | Aimée T. Classen, Conceptualization, Funding acquisition, Supervision, Writing – review and editing | David W. Inouye, Conceptualization, Funding acquisition, Supervision, Writing – review and editing | Ruth Simberloff, Data curation, Investigation, Methodology, Writing – review and editing | Patrick O. Sorensen, Conceptualization, Funding acquisition, Writing – review and editing | William Thomas, IV, Data

curation, Investigation, Methodology, Writing – review and editing | Jennifer A. Rudgers, Conceptualization, Funding acquisition, Supervision, Writing – review and editing | Stephanie N. Kivlin, Conceptualization, Data curation, Funding acquisition, Investigation, Methodology, Project administration, Resources, Supervision, Writing – original draft, Writing – review and editing

## DATA AVAILABILITY

The data that support the findings of this study are openly available. AMF fungal SSU sequences are archived under NCBI SRA, BioProject PRJNA1305024. Raw data, assemblies, and annotations for metagenome data are available through the JGI's data portal under the IMG Genome IDs and Gold Analysis project IDs listed in Table S6. Other data and R code are available at https://github.com/hbbshulman/KivlinLab_2018RMBL_gradients.

## ADDITIONAL FILES

The following material is available online.

### Supplemental Material

**Supplemental material (mSystems00523-25-S0001.pdf).** Figures S1-S3, plus additional methods related to sequencing and bioinformatics.
**Supplemental Tables (mSystems00523-25-S0002.xlsx).** Tables S1-S6.

### Open Peer Review

**PEER REVIEW HISTORY (review-history.pdf).** An accounting of the reviewer comments and feedback.

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
