## [Reviewer comments · mSystems]

Nutrient limitation shapes functional traits of mycorrhizal fungi & phosphorus cycling bacteria across an elevation gradient

Hannah Shulman, Jessica Pyle, Aimée Classen, David Inouye, Ruth Simberloff, Patrick Sorensen, William Thomas, Jennifer Rudgers, and Stephanie Kivlin

Corresponding Author(s): Hannah Shulman, The University of Tennessee System

Review Timeline:

Submission Date:	April 16, 2025
Editorial Decision:	July 18, 2025
Revision Received:	September 29, 2025
Accepted:	October 15, 2025

Editor: Leonora Bittleston

Reviewer(s): Disclosure of reviewer identity is with reference to reviewer comments included in decision letter(s). The following individuals involved in review of your submission have agreed to reveal their identity: Mia Rose Maltz (Reviewer #1)

Transaction Report:

DOI: <https://doi.org/10.1128/msystems.00523-25>

Re: mSystems00523-25 (Nutrient limitation shapes functional traits of mycorrhizal fungi & phosphorus cycling bacteria across an elevation gradient)

Dear Dr. Hannah B Shulman:

Thank you for submitted your work to mSystems. Below you will find my comments, instructions from the mSystems editorial office, and the reviewer comments.

First, I apologize for how long the review process has taken. All of the suggested reviewers declined, and I had a difficult time securing reviewers. But now we have two expert reviews and both reviewers had positive comments about the study. Both also noted some concerns, which would need to be addressed prior to publication.

Revision Guidelines

Sincerely,
Leonora Bittleston
Editor
mSystems

Reviewer #1 (Comments for the Author):

This is an innovative and thoughtful paper that leverages ecological theory to link biogeochemical cycling to fungal traits and microbial interactions. This study provides evidence of trade offs in phosphorus cycling strategies across elevation, enhancing predictive power for mountaintop ecosystems faced with climate change. This study was methodologically sound and well

written, with the exception of minor errors which are easily correctable. Hypotheses were well thought out with appropriate rationale to address how community dynamics shift with elevation and changing P concentrations.

Quantifying aseptate AM fungal hyphae should also take into account the branching angle of hyphae. Was this addressed in Ref. 55, as per line 180?

The footnote, after line 145, while informative, would be good as a Box or conceptual figure as the format of footnote was off-putting, unless that is customary for mSystems.

Assumptions made about PCBs were based primarily on shotgun metagenomics, which these data (gene products related to p-cycling, e.g., p-cycling genes, protein-encoding genes) were analyzed well using novel tools and relying on appropriate databases. Yet, ancillary targeted amplicon sequencing of 16S marker genes would have provided additional value, such as to substantiate the PCB community, as paired with the AM fungal community, and thus would have added value to the paper, along with the small subunit rDNA NS31/AML2 amplicon data set.

Although interesting, L293-295 should be moved from Results section to the Discussion section and further drawn out: "These results suggest that fungal-correlated shifts in PCB composition may be functionally redundant, related to niche structuring, and potentially represent local co-adaptation of bacteria and mycorrhizal fungi to soil conditions like soil carbon dynamics";

same comment 297-98: "indicating that the observed AMF-PCB relationship is unique to the P-cycling bacterial community, and not a characteristic of total bacterial community structure; same comment L 311-12 "suggesting that community filtering differs among P-cycling clades.";

same comment L337-39: "These results indicate that elevational shifts in AM fungal diversity and nutrient availability are related to the function, composition, and structure of fungal-bacterial communities.";

L353-355: make explicit if aseptate AM fungi were quantified as hyphal length, how ample P might reduce investment into extraradical hyphae, here: "suggest that rhizophilic AM fungi, which preferentially allocate growth to the intraradical space, are associated with less total hyphal colonization of the soil and less phosphorus limitation."

Although beyond the scope of the study, how would one test predictions of ancestral-type hyphae being longer-lived, as you suggest, would you suggest C to N ratios? Draw this out and bring in more evidence from ectomycorrhizal, soil science, and saprotrophic fungal literature and mycorrhizal exploration types "...and their mechanism of stress tolerance could be decomposing or otherwise turning over more slowly than other mycorrhizal types, therefore demanding less C from host plants and returning more P per C in resource-limited environments."

Minor changes detailed here:

on line 242: principle coordinate axes should be corrected as principal coordinate axes;

line 263: correct "or" to "nor" Neither PO43- or hyphal lengths;

line 330: add an "f" to ungal abundance;

line 355-6: sentence fragment (While ancestral AM fungi were correlated with more total hyphal colonization of the soil and high phosphorus limitation.);

line 359 capitalization, lowercase "it" for "But It seems contradictory... ";

line 368: fix citations.;

line 405: fix hyposphere to hyphosphere;

line 431 correct "our find" to "our findings";

L396: What would the mechanism be for "... greater functional diversity among AM fungi may enhance bacterial community assembly and phosphorus cycling capacity."?

Reviewer #2 (Comments for the Author):

The manuscript "Nutrient limitation shapes functional traits of mycorrhizal fungi & phosphorus cycling bacteria across an elevation gradient" uses soil sampling from high-elevation ecosystems to explore connections between arbuscular mycorrhizal fungi and phosphorus cycling bacteria. The study's results show clear patterns with elevational change, where higher elevation soils were dominated by stress-tolerant fungi and mineral P-solubilizing bacteria while lower elevation soils (with more available P) were dominated by root-colonizing fungi and organic P-mineralizing bacteria. In particular, I really appreciated how they sampled across 3 different mountains for replication. The results are interesting and I think they are relevant across multiple fields of study.

I had a couple of minor concerns: (1) it was not clear why the network was done in the way the authors chose to do it, and what it adds to the study. There were very few results or details from the network included, and no discussion of the many limitations from correlational approaches with relative abundance data. I would like to see a more thorough approach, with additional discussion and integration into the rest of the manuscript. (2) There was not enough information to verify that the authors

checked to make sure that linear models were appropriate for the data in the alpha diversity analyses, and also if they controlled for multiple testing where necessary.

Overall, the manuscript is well-written and presents compelling evidence of relevant bacterial-fungal functional patterns.

Specific comments:

Line 120: Looks like parts of a reference remain from when formatting was changed. There is a similar issue on Line 370.

Line 130: I think this footnote could be incorporated into the main text of the introduction

Line 201: how can the authors be sure that all the P-cycling genes in the metagenomes were of bacterial origin? On lines 212-214 the text mentions summarizing contigs by species identity, so maybe this is how? Or maybe authors are only mining genes of bacterial origin? If so, why not look into fungal genes? The approach and reasoning should be made clearer in the text.

Line 235: The symbol didn't come through properly here and on many other lines in the manuscript (e.g. line 259, 261, 266, 271, 273)

Line 249: Why use Kendall rank correlation, instead of an approach that might be more suited for relative abundance data?

Line 276-292: How much of the total variation was explained by the dbRDA (constrained portion)? Often this is low in ecological studies, and it would be good to report it.

Line 298: Switches to present tense here.

Line 440: I think this should say Hypothesis 3 here instead of Hypothesis 2

This is an innovative and thoughtful paper that leverages ecological theory to link biogeochemical cycling to fungal traits and microbial interactions. This study provides evidence of tradeoffs in phosphorus cycling strategies across elevation, enhancing predictive power for mountaintop ecosystems faced with climate change.

This study was methodologically sound and well written, with the exception of minor errors which are easily correctable. Hypotheses were well thought out with appropriate rationale to address how community dynamics shift with elevation and changing P concentrations. Suggestions for improvement below:

Quantifying aseptate AM fungal hyphae should also take into account the branching angle of hyphae. Was this addressed in Ref. 55, as per line 180? The footnote, after line 145, while informative, would be good as a Box or conceptual figure as the format of footnote was off-putting, unless that is customary for mSystems. Assumptions made about PCBs were based primarily on shotgun metagenomics, which these data (gene products related to p-cycling, e.g., p-cycling genes, protein-encoding genes) were analyzed well using novel tools and relying on appropriate databases. Yet, ancillary targeted amplicon sequencing of 16S marker genes would have provided additional value, such as to substantiate the PCB community, as paired with the AM fungal community, and thus would have added value to the paper, along with the small subunit rDNA NS31/AML2 amplicon data set. Although interesting, L293-295 should be moved from Results section to the Discussion section and further drawn out: "These results suggest that fungal-correlated shifts in PCB composition may be functionally redundant, related to niche structuring, and potentially represent local co-adaptation of bacteria and mycorrhizal fungi to soil conditions like soil carbon dynamics"; same comment 297-98: "indicating that the observed AMF-PCB relationship is unique to the P-cycling bacterial community, and not a characteristic of total bacterial community structure; same comment L 311-12 "suggesting that community filtering differs among P-cycling clades."; same comment L337-39: "These results indicate that elevational shifts in AM fungal diversity and nutrient availability are related to the function, composition, and structure of fungal-bacterial communities."; L353-355: make explicit if aseptate AM fungi were quantified as hyphal length, how ample P might reduce investment into extraradical hyphae, here: "suggest that rhizophilic AM fungi, which preferentially allocate growth to the intraradical space, are associated with less total hyphal colonization of the soil and less phosphorus limitation." Although beyond the scope of the study, how would one test predictions of ancestral-type hyphae being longer-lived, as you suggest, would you suggest C to N ratios? Draw this out and bring in more evidence from ectomycorrhizal, soil science, and saprotrophic fungal literature and mycorrhizal exploration types "...and their mechanism of stress tolerance could be decomposing or otherwise turning over

more slowly than other mycorrhizal types, therefore demanding less C from host plants and returning more P per C in resource-limited environments."

Minor changes detailed here:

on line 242: principle coordinate axes should be corrected as principal coordinate axes;

line 263: correct "or" to "nor" Neither PO₄³⁻ or hyphal lengths;

line 330: add an "f" to ungal abundance ;

line 355-6: sentence fragment (While ancestral AM fungi were correlated with more total hyphal colonization of the soil and high phosphorus limitation.);

line 359 capitalization, lowercase "it" for "But It seems contradictory... ";

line 368: fix citations.;

line 405: fix hyposphere to hyphosphere;

line 431 correct "our find" to "our findings";

L396: What would the mechanism be for "... greater functional diversity among AM fungi may enhance bacterial community assembly and phosphorus cycling capacity."?

Reviewer Comments in Bold, Author Responses in Regular Text

Reviewer #1 (Comments for the Author):

This is an innovative and thoughtful paper that leverages ecological theory to link biogeochemical cycling to fungal traits and microbial interactions. This study provides evidence of trade offs in phosphorus cycling strategies across elevation, enhancing predictive power for mountaintop ecosystems faced with climate change. This study was methodologically sound and well written, with the exception of minor errors which are easily correctable. Hypotheses were well thought out with appropriate rationale to address how community dynamics shift with elevation and changing P concentrations.

We thank the reviewer for their positive and encouraging assessment of our work. We are glad the application of ecological theory to link fungal traits, microbial interactions and phosphorus cycling was seen as innovative and methodologically sound. We appreciate the recognition of our hypotheses and rationale and have addressed all suggested revisions to improve clarity and accuracy.

Quantifying aseptate AM fungal hyphae should also take into account the branching angle of hyphae. Was this addressed in Ref. 55, as per line 180?

While incorporating hyphal branching angle would definitely strengthen the trait-based framework we apply in this study, this information is not available with the gridline-intercept method we used (as described in Sylvia 1992) Specifically, the dispersion of soil in sodium hexametaphosphate breaks up soil aggregates and frees hyphae for quantification, but also disrupts the spatial orientation and structure of the hyphal network, including the branching angles. While we can reliably quantify aseptate hyphae length using this method, we can't retain morphological features such as branching angle during sample preparation.

Sylvia, D. M. (1992). Quantification of external hyphae of vesicular-arbuscular mycorrhizal fungi. In *Methods in Microbiology* (Vol. 24, pp. 53–65). Elsevier.
[https://doi.org/10.1016/s0580-9517\(08\)70086-2](https://doi.org/10.1016/s0580-9517(08)70086-2)

The footnote, after line 145, while informative, would be good as a Box or conceptual figure as the format of footnote was off-putting, unless that is customary for mSystems.

We have incorporated key points from the footnote into the main text (see L130-135) and removed the footnote.

Assumptions made about PCBs were based primarily on shotgun metagenomics, which these data (gene products related to p-cycling, e.g., p-cycling genes, protein-encoding genes) were analyzed well using novel tools and relying on appropriate databases. Yet, ancillary targeted amplicon sequencing of 16S marker genes would have provided additional value, such as to substantiate the PCB community, as paired with the AM

fungal community, and thus would have added value to the paper, along with the small subunit rDNA NS31/AML2 amplicon data set.

Thank you for raising this issue. While including a 16S rRNA gene amplicon dataset could provide complementary information, our study design aimed to minimize between-dataset biases by analyzing prokaryotic communities from a unified shotgun metagenomic dataset. Introducing a separate amplicon sequencing effort would have added complexity and instrumentation error. To account for overall bacterial community structure, we did analyze the single-copy *rpoB* marker gene extracted from the shotgun metagenomes, which allowed us to evaluate broad taxonomic patterns alongside functional gene inventories and AM fungal SSU amplicon data. The AM fungal amplicon dataset in this case was necessary as AM fungal DNA reads were not well represented in the shotgun dataset (L206-208), while bacterial community structure can be assessed from shotgun metagenome data.

Ogier, J.-C., Pagès, S., Galan, M., Barret, M., & Gaudriault, S. (2019). *rpoB*, a promising marker for analyzing the diversity of bacterial communities by amplicon sequencing. *BMC Microbiology*, 19(1), 171. <https://doi.org/10.1186/s12866-019-1546-z>

Durand, K., Ogier, J.-C., & Nam, K. (2025). The evaluation of shotgun sequencing and *rpoB* metabarcoding for taxonomic profiling of bacterial communities. *BMC Microbiology*, 25(1), 413. <https://doi.org/10.1186/s12866-025-04149-3>

Although interesting, L293-295 should be moved from Results section to the Discussion section and further drawn out: "These results suggest that fungal-correlated shifts in PCB composition may be functionally redundant, related to niche structuring, and potentially represent local co-adaptation of bacteria and mycorrhizal fungi to soil conditions like soil carbon dynamics";

same comment 297-98: "indicating that the observed AMF-PCB relationship is unique to the P-cycling bacterial community, and not a characteristic of total bacterial community structure; same comment L 311-12 "suggesting that community filtering differs among P-cycling clades.";

same comment L337-39: "These results indicate that elevational shifts in AM fungal diversity and nutrient availability are related to the function, composition, and structure of fungal-bacterial communities.";

We appreciate these suggestions to move interpretive statements from the results to the discussion. Our intent in including brief interpretations in these sections was to aid readers who may be less familiar with joint fungal–bacterial analyses by providing immediate context for the statistical outcomes. We view these short interpretive bridges as essential for comprehension, particularly when describing these novel interkingdom interactions between AM fungi and P-cycling bacteria.

We have ensured that these statements remain concise, descriptive, and directly tied to the presented results, while fuller interpretation and broader ecological implications are expanded upon in the discussion. We believe this structure improves readability for a diverse audience while maintaining the standard separation between results and discussion.

L353-355: make explicit if aseptate AM fungi were quantified as hyphal length, how ample P might reduce investment into extraradical hyphae, here: "suggest that rhizophilic AM fungi, which preferentially allocate growth to the intraradical space, are associated with less total hyphal colonization of the soil and less phosphorus limitation."

We appreciate this helpful feedback and have revised this paragraph to emphasize hyphal quantification method, results, and implications for P-limitation effects, see L356-363:

While we were not able to determine species-specific hyphal growth patterns, the taxonomically based functional groups of the ERA framework (Weber et al. 2019) and hyphal data taken together show that hyphal allocation traits track our measurements of hyphal length density in bulk soil. Rhizophilic AM fungi, which preferentially allocate growth to the intraradical space, were associated with less hyphal colonization of bulk soil and less P limitation. Ancestral AM fungi were correlated with more total hyphal colonization of the soil and high P limitation. This pattern supports that ample P availability reduces selective pressure for extraradical foraging, leading to lower investment in soil-exploring hyphae.

Although beyond the scope of the study, how would one test predictions of ancestral-type hyphae being longer-lived, as you suggest, would you suggest C to N ratios? Draw this out and bring in more evidence from ectomycorrhizal, soil science, and saprotrophic fungal literature and mycorrhizal exploration types "...and their mechanism of stress tolerance could be decomposing or otherwise turning over more slowly than other mycorrhizal types, therefore demanding less C from host plants and returning more P per C in resource-limited environments."

We appreciate the suggestion to elaborate on testing predictions regarding the longevity of ancestral hyphae and to integrate evidence from ectomycorrhizal, soil science, and saprotrophic fungal literature. See L375-378:

*Rhizophilic taxa like *Glomus* and *Rhizophagus* have high hyphal turnover rates, fast growth rates, and cell walls made up of more acid-hydrolysable compounds compared to other fungi, which could result in less hyphae being observed in the soil due to faster decomposition (Chagnon et al. 2013; Staddon et al. 2003). Future research focusing on species-specific differences in AM fungal turnover and hyphal chemistry, e.g. C to N ratios, would improve understanding of different strategies for regulating host C supply as it has for ectomycorrhizas and saprotrophs (74, 75).*

Minor changes detailed here:

on line 242: principle coordinate axes should be corrected as principal coordinate axes;
line 263: correct "or" to "nor" Neither PO₄³⁻ or hyphal lengths;
line 330: add an "f" to ungal abundance;
line 355-6: sentence fragment (While ancestral AM fungi were correlated with more total hyphal colonization of the soil and high phosphorus limitation.);
line 359 capitalization, lowercase "it" for "But It seems contradictory... ";
line 368: fix citations.;
line 405: fix hyposphere to hyphosphere;
line 431 correct "our find" to "our findings";

We have corrected all these minor errors, thank you for your attention to detail.

L396: What would the mechanism be for "... greater functional diversity among AM fungi may enhance bacterial community assembly and phosphorus cycling capacity."?

See added discussion text on 403-407:

We also observed that the diversity of AM fungal functional groups is positively related to their interactions with PCBs, suggesting that greater functional diversity among AM fungi may shift bacterial community assembly for enhanced P cycling capacity. This may occur if diversity in hyphal growth strategies create a broader range of nutrient niches and soil microhabitats, fostering complementary PCB functions and enhancing overall phosphorus turnover.

We appreciate the reviewer's thoughtful feedback and constructive suggestions, which have helped us refine our interpretations and improve the clarity of the manuscript. We believe these revisions strengthen the work and enhance its contribution to understanding fungal–bacterial P cycling.

Reviewer #2 (Comments for the Author):

The manuscript "Nutrient limitation shapes functional traits of mycorrhizal fungi & phosphorus cycling bacteria across an elevation gradient" uses soil sampling from high-elevation ecosystems to explore connections between arbuscular mycorrhizal fungi and phosphorus cycling bacteria. The study's results show clear patterns with elevational change, where higher elevation soils were dominated by stress-tolerant fungi and mineral P-solubilizing bacteria while lower elevation soils (with more available P) were dominated by root-colonizing fungi and organic P-mineralizing bacteria. In particular, I really appreciated how they sampled across 3 different mountains for replication. The results are interesting and I think they are relevant across multiple fields of study

We thank the reviewer for their thoughtful and encouraging comments. We are pleased that the elevational patterns we observed and our replication across three mountains were appreciated. We agree the results have relevance across multiple fields and have addressed all suggested revisions to strengthen the manuscript.

I had a couple of minor concerns: (1) it was not clear why the network was done in the way the authors chose to do it, and what it adds to the study. There were very few results or details from the network included, and no discussion of the many limitations from correlational approaches with relative abundance data. I would like to see a more thorough approach, with additional discussion and integration into the rest of the manuscript.

Thank you for raising this question! We started with the compositional analysis (dbRDA & variance partitioning, Fig 2 and Supplemental Table 1) to determine the conditioning effects of the AM fungal community versus the edaphic/environmental factors on the phosphorus-cycling bacterial community. After finding that there was a distinct conditioning effect of AM fungi on the PCBs, our goal was to then use the co-occurrence network to explore potential associations between specific AM fungal taxa and p-cycling functional gene groups, moving beyond overall community structure to identify specific guilds or taxa that were consistently associated with particular functional traits. We have added information to better explain the purpose of the network (L334-336). While earlier drafts of this manuscript contained figures of the networks themselves, we made the choice to remove them for clarity and simplicity, as we believe Fig 4B efficiently conveys the key findings relevant to this story.

We understand and share your concern about limitations using relative abundance data. We center-log ratio normalized our datasets to handle the compositional structure of the data, minimizing the issue of closed-sums for compositional and co-occurrence analysis. We took an extra measure to reduce this error in the shotgun metagenome dataset by normalizing gene copies to the rpoB gene before clr transformation.

Gloor, G. B., Macklaim, J. M., Pawlowsky-Glahn, V., & Egozcue, J. J. (2017). Microbiome Datasets Are Compositional: And This Is Not Optional. *Frontiers in Microbiology*, 8, 2224. <https://doi.org/10.3389/fmicb.2017.02224>

Brunner, J., Robinson, A. J., & Chain, P. S. G. (2023). Combining compositional data sets introduces error in covariance network reconstruction. *ISME Communications*, 4(1), ycae057. <https://doi.org/10.1093/ismeco/ycae057>

2) There was not enough information to verify that the authors checked to make sure that linear models were appropriate for the data in the alpha diversity analyses, and also if they controlled for multiple testing where necessary. Overall, the manuscript is well-written and presents compelling evidence of relevant bacterial-fungal functional patterns.

Methods have been updated (L232) to reflect that we verified the assumptions of linear regression for each AMF diversity response. Model residuals exhibited homoscedastic scatter around zero in residual–fitted plots and no strong deviations from normality, supporting the use of linear models for these data.

We report raw p-values rather than adjusted values (e.g., Bonferroni correction). We made this decision because:

- a) The predictors for each model were selected a priori based on our hypotheses about drivers of microbial phosphorus cycling.
- b) While we tested multiple response variables, each response was analyzed once using a single mixed-effects model in which all predictors were evaluated jointly. Because predictors are tested within each model and model selection uses AIC rather than large numbers of independent null-hypothesis tests, it is not necessary to apply multiple-testing corrections.

Specific comments:

Line 120: Looks like parts of a reference remain from when formatting was changed. There is a similar issue on Line 370.

We have fixed the in-text citation formatting in both these instances, thank you.

Line 130: I think this footnote could be incorporated into the main text of the introduction

We have incorporated key points from the footnote into the main text (L130-135) and removed the footnote.

Line 201: how can the authors be sure that all the P-cycling genes in the metagenomes were of bacterial origin? On lines 212-214 the text mentions summarizing contigs by species identity, so maybe this is how? Or maybe authors are only mining genes of bacterial origin? If so, why not look into fungal genes? The approach and reasoning should be made clearer in the text.

Thank you for raising this point. The method of taxonomic identification for functional genes is described in the supplemental bioinformatics appendix. With our bioinformatics pipeline, the taxonomic breakdown of P-cycling contigs was 98.4 % Bacteria, 0.33% Archaea, 0.2 % fungi, 0.24% other eukaryotes, and 0.84% unassigned. Therefore, we focused on phosphorus-cycling bacteria in the metagenome data. This was as expected, fungi are usually poorly represented in shotgun metagenome datasets from bulk soil due to combined issues of abundance of fungal DNA in the soil and database bias, see Werbin et al 2025. We appreciate your concern and have updated our methods to better explain our approach on L207.

Werbin, Z., Dukovski, I., Mankel, D., Anthony, W. E., Segrè, D., Bhatnagar, J. M., & Felici, M. (2025). Improved detection of fungi and uncultivated microorganisms in soil metagenomes using a comprehensive genome database. In *bioRxiv* (p. 2025.03.21.644662). <https://doi.org/10.1101/2025.03.21.644662>

Line 235: The symbol didn't come through properly here and on many other lines in the manuscript (e.g. line 259, 261, 266, 271, 273)

Thank you, we'll discuss this with the editors.

Line 249: Why use Kendall rank correlation, instead of an approach that might be more suited for relative abundance data?

We chose to use Kendall rank correlation for the network analysis to accommodate the random effect structure inherent to our design. We selected Kendall rank correlation over Spearman rank correlation or Pearson correlation because its rank-based, pairwise approach is more robust to the non-normality, and zero inflation in relative abundance data. We also reduced sparsity in the Kendall rank networks by binning taxa and genes at higher taxonomic and functional levels, and applied CLR transformation prior to analysis as described above.

Graphical models like SPIEC-EASI are definitely better at handling the compositional nature of relative abundance data. However, our goal was to generate separate networks for each elevation and mountain peak to investigate regional variation in community co-occurrence patterns. This approach was more tractable using pairwise Kendall correlations, which allowed us to preserve local-scale ecological structure across groups. Put simply: we didn't have the sample size to make a more sophisticated network at each elevation on each mountain, and choose to prioritize more robust replication of networks over a single, robust network.

Line 276-292: How much of the total variation was explained by the dbRDA (constrained portion)? Often this is low in ecological studies, and it would be good to report it.

These percentages are in the axis labels in Fig 2A-B, and we have also added the residuals of the variance partitioning to Fig 2C-D.

Line 298: Switches to present tense here.

Fixed, thank you!

Line 440: I think this should say Hypothesis 3 here instead of Hypothesis 2

We only have two hypotheses in the manuscript, so the reference at line 440 is correct as “Hypothesis 2.” We have double-checked the numbering throughout to ensure consistency.

Thank you for your thoughtful and constructive feedback, it has substantially improved the clarity and rigor of our manuscript! We believe the revisions have addressed all comments and strengthened the work, and we thank the reviewer again for their time and expertise.

Re: mSystems00523-25R1 (Nutrient limitation shapes functional traits of mycorrhizal fungi & phosphorus cycling bacteria across an elevation gradient)

Dear Dr. Hannah B Shulman:

I appreciate the changes you made in response to the reviews and I think this is a very nice comparative study.

Your manuscript has been accepted, and I am forwarding it to the ASM production staff for publication. Your paper will first be checked to make sure all elements meet the technical requirements. ASM staff will contact you if anything needs to be revised before copyediting and production can begin. Otherwise, you will be notified when your proofs are ready to be viewed.

Sincerely,
Leonora Bittleston
Editor
mSystems